# A Novel Reverse Transcription Loop-Mediated Isothermal Amplification Method for Rapid Detection of SARS-CoV-2

**DOI:** 10.3390/ijms21082826

**Published:** 2020-04-18

**Authors:** Renfei Lu, Xiuming Wu, Zhenzhou Wan, Yingxue Li, Xia Jin, Chiyu Zhang

**Affiliations:** 1Clinical Laboratory, Nantong Third Hospital Affiliated to Nantong University, Nantong 226006, China; rainman78@163.com; 2Pathogen Discovery and Evolution Unit, Institut Pasteur of Shanghai, Chinese Academy of Sciences, Shanghai 200031, China; xmwu@ips.ac.cn (X.W.); yxli@ips.ac.cn (Y.L.); 3Medical Laboratory of Taizhou Fourth People’s Hospital, Taizhou 225300 China; wanlv@126.com; 4Shanghai Public Health Clinical Center, Fudan University, Jinshan District, Shanghai 201508, China; jinxia@shphc.org.cn

**Keywords:** COVID-19, SARS-CoV-2, POCT, LAMP, pneumonia

## Abstract

COVID-19 has become a major global public health burden, currently causing a rapidly growing number of infections and significant morbidity and mortality around the world. Early detection with fast and sensitive assays and timely intervention are crucial for interrupting the spread of the COVID-19 virus (SARS-CoV-2). Using a mismatch-tolerant amplification technique, we developed a simple, rapid, sensitive and visual reverse transcription loop-mediated isothermal amplification (RT-LAMP) assay for SARS-CoV-2 detection based on its *N* gene. The assay has a high specificity and sensitivity, and robust reproducibility, and its results can be monitored using a real-time PCR machine or visualized via colorimetric change from red to yellow. The limit of detection (LOD) of the assay is 118.6 copies of SARS-CoV-2 RNA per 25 μL reaction. The reaction can be completed within 30 min for real-time fluorescence monitoring, or 40 min for visual detection when the template input is more than 200 copies per 25 μL reaction. To evaluate the viability of the assay, a comparison between the RT-LAMP and a commercial RT-qPCR assay was made using 56 clinical samples. The SARS-CoV-2 RT-LAMP assay showed perfect agreement in detection with the RT-qPCR assay. The newly-developed SARS-CoV-2 RT-LAMP assay is a simple and rapid method for COVID-19 surveillance.

## 1. Introduction

Since December 2019, an emerging infectious disease (COVID-19), caused by the novel coronavirus SARS-CoV-2, has emerged in Wuhan, China [1,2,3]. SARS-CoV-2 is the seventh coronavirus that causes human infections. Like SARS-CoV and MERS-CoV, SARS-CoV-2 can lead to lethal pneumonia, but it also has a stronger human-to-human transmission capacity than SARS-CoV and MERS-CoV [4,5]. As of 1 April 2020, it has caused 876,898 infections in 203 countries, including 43,477 deaths. The on-going COVID-19 pandemic is a new and huge global public health threat [6,7]. In the absence of suitable antiviral drugs or vaccines, simple, rapid and reliable detection of the SARS-CoV-2 infection is critical not only for the prevention and control of the COVID-19 pandemic, but also for clinical treatment [8]. 

Real-time PCR (qPCR) is the most robust and widely used technology for the qualitative and quantitative diagnosis of viruses, including coronaviruses [8,9,10,11]. Since the outbreak of COVID-19, many real-time qPCR (RT-qPCR) assays had been developed, and have played an essential role in laboratory confirmation of SARS-CoV-2 infection [3,9,12]. However, the RT-qPCR assay requires sophisticated equipment and highly trained personnel, and it is relatively time-consuming (about 1.5–2 h), which limits its capacity to meet the demand for detecting the virus in a rapidly growing number of patients with COVID-19 infection, suspected infection, or close contact with confirmed cases. Therefore, fast, simple, and sensitive point-of-care testing (POCT) assays are urgently needed to facilitate the detection of SARS-CoV-2 infection. 

LAMP is a simple, fast (within 50 min) and sensitive isothermal amplification technique, that is less dependent on sophisticated equipment, and has been widely used for the development of POCT techniques for the detection of viruses and other pathogens in field and/or resource-limited settings [13]. Recently, we developed a mismatch-tolerant version of the LAMP method that has higher sensitivity and a faster reaction speed than the conventional ones [14,15]. In this study, we applied the mismatch-tolerant technique to develop novel real-time fluorescent and visual RT-LAMP assays for the rapid and sensitive detection of SARS-CoV-2 RNA, and evaluated the novel assays using clinical samples.

## 2. Results

### 2.1. Primer Design

SARS-CoV-2 viruses are relatively conserved with very low sequence divergence [16]. To develop RT-LAMP assays for SARS-CoV-2 detection, we aligned the SARS-CoV-2 genomic sequence with those of six other human CoVs, including SARS-CoV, MERS-CoV, HCoV-HKU-1, HCoV-NL63, HCoV-OC43 and HCoV-229E. Several sets of SARS-CoV-2-specific LAMP primers, targeting *N*, *S* and *RdRp* genes, were designed according to the SARS-CoV-2 genomic sequence, using the open access Primer Explorer V.5 software tool (http://primerexplorer.jp/). The primer specificity was first evaluated via sequence alignment with the other six human coronaviruses and the performance of a homology search using the BLAST tool in NCBI, and then by performing an RT-LAMP reaction with non-template control (NTC). After excluding the primer sets with non-specific amplification in the NTC reaction, one primer set targeting the N gene was found to have higher amplification efficiency, and was thus selected for use in the development of the method (Table 1 and Figure 1).

### 2.2. Specificity and Sensitivity of the SARS-CoV-2 RT-LAMP Assay

Specificity tests showed that there were no amplification curves or very weak amplification signals observed for all 17 respiratory viruses after 50 min of reaction, indicating the high specificity of the assay (Figure 2). Because corresponding clinical samples or standard strains are unavailable for SARS-CoV and MERS-CoV, we did not test the specificity for the two coronaviruses. However, the sequence comparison suggests that the RT-LAMP may also amplify SARS-CoV due to high sequence similarity (Figure 1b).

Ten-fold serial dilutions of SARS-CoV-2 RNA standard, from 10^5^ to 10^0^ copies per μL, were used to determine the sensitivity of the RT-LAMP assay. RNA inputs of 2 × 10^5^ to 2 × 10^1^ copies per 25 μL reaction generated typical S-shaped amplification curves, whereas the RNA input of 2 copies did not yield an obvious amplification curve (Figure 3a), indicating that the RT-LAMP can detect as few as 20 copies per reaction. Importantly, all amplification curves appeared within 15 min and entered a plateau phase within 20 min when the template input was more than 200 copies (Figure 3a), indicating that the assay is fast. 

Sensitivity testing showed that when the template input was more than 393 copies of SARS-CoV-2 RNA, all 10 reactions (100%) displayed positive amplification. When the template input was 79 and 16 copies, eight and six of the 10 reaction replicates showed positive, respectively (Table 2). The limit of detection (LOD) of the new RT-LAMP assay was calculated to be 118.6 copies per reaction.

### 2.3. Visual Detection

For convenient use, a visual detection version of the SARS-CoV-2 RT-LAMP assay was developed. The reaction gave clear color change, from red to yellow, for all samples tested for over 40 min (40–60 min). Therefore, the 40 min time period was selected as the optimal protocol (cut-off) for the visual assays (Figure 3b). A very faint color change was observed when 20 copies of the RNA standard were inputted. These results indicated that the colorimetric reaction system’s sensitivity is similar to that of the fluorescent detection system (Figure 3).

### 2.4. Evaluation of the Novel SARS-CoV-2 RT-LAMP Assay Using Clinical Samples

To evaluate the clinical application of the novel RT-LAMP assay, a comparison with a commercial RT-qPCR assay was performed, using 56 clinical samples collected from both COVID-19-suspected patients and control populations. Of these samples, 34 and 18 were detected as SARS-CoV-2 positive and negative by both assays, respectively (Table 3). The concordance rate between both assays was 92.9%. Two samples were detected as positive by one assay, but negative by another. The two negative samples detected by the novel RT-LAMP assay had high threshold cycle (CT) values (>38) in the RT-qPCR assay, indicating a very low viral copy number.

## 3. Discussion

COVID-19 is a newly emerging, life threatening respiratory disease caused by SARS-CoV-2, a newly identified human coronavirus [1,2,3,7]. As the seventh human coronavirus and the third most lethal coronavirus, SARS-CoV-2 has a higher affinity to human receptor ACE2 than SARS-CoV [4,5], and a greater human-to-human transmission capacity than other human coronaviruses [17,18]. Because of its newness, communicability, and rapid spread, SARS-CoV-2 has led to a global pandemic and created great international concern. As of 1 April 2020, 81,691 people in China and 798,200 people in over 200 other countries have been infected with SARS-CoV-2, and a total of 44,036 individuals have died from COVID-19. 

Transmission of SARS-CoV-2 appears to mainly occur in the early stage of infection, with higher viral loads in the nasopharyngeal tract, or soon after the symptom-onset of COVID-19 [19,20]. Early diagnosis, and the timely implementation of intervention and quarantine measures, are crucial to prevent the spread of the virus and optimize clinical management. Therefore, the development of simple, rapid and reliable diagnostic assays is of high priority for the prevention and control of COVID-19. Many RT-qPCR assays have been developed since the outbreak, and contributed to the confirmation of most SARS-CoV-2 infections during the pandemic in China [3,9,12]. Currently, although the COVID-19 pandemic has been contained following the strong public health intervention efforts of the Chinese government, the virus is spreading quickly outside China, including into some resource-limited areas. The stringent requirements of facilities and professional workers, and its time-consuming nature, limit the use of RT-qPCR assays in some undeveloped countries. Furthermore, the presence of asymptomatic infections and the long incubation period of COVID-19 increase the difficulty of diagnosis, because some SARS-CoV-2 carriers may not visit the hospital for diagnosis due to lack of symptoms or signs of infection [21]. Therefore, simple, rapid and sensitive POCT detection assays, that can be deployed more widely, will be very helpful for the early diagnosis of SARS-CoV-2 infection, and should be developed.

One promising POCT method, LAMP, has been used for the detection of various pathogens because of its high sensitivity, rapid reaction speed, relatively simple operation, and visual determination capability [14,15,22,23]. We recently upgraded the LAMP method to a mismatch-tolerant version [14,15]. Compared to the conventional LAMP method, the novel version contains an additional 0.15 U of high-fidelity DNA polymerase, which confers upon it a higher applicability to highly variable viruses, and a 10–15 min faster reaction speed. In this study, we established a rapid and sensitive one-step single-tube RT-LAMP assay, for the detection of SARS-CoV-2 using the mismatch-tolerant technique. The assay can be performed at 63 °C for 50 min in a real-time PCR instrument, for real-time monitoring using fluorescent dye, or for 40 min in a regular PCR machine or heating block (e.g., dry incubator or water bath), for visual detection using pH-sensitive indicator dyes. In the real-time monitoring system, SYTO9 is used as a fluorescent dye which has a minimal inhibitory effect on LAMP amplification [10,14]. In the visual detection system, cresol red is used as a pH indicator dye because of a clear color contrast between negative (red) and positive (yellow) reactions after 40 min of incubation at 63 °C; a color change from red to orange or yellow is defined as SARS-CoV-2 positive [24]. 

The novel RT-LAMP assay has a high sensitivity, with a LOD of 118.6 copies per reaction, and shows no cross-reactivity with 17 common human respiratory viruses, including four other human coronaviruses (OC43, 229E, HKU-1 and NL63). In particular, when the template input is more than the LOD, the assay has a very fast detection speed of less than 20 min. The robustness of the novel assay was tested using 56 clinical samples from COVID-19 patients and control populations in Nantong. The novel RT-LAMP assay showed a high consistency (92.9%) with a commercial RT-qPCR assay for SARS-CoV-2 RNA detection. There were four samples showing inconsistent results by both the RT-qPCR assay and the novel RT-LAMP assay. The reason for two samples testing positive by the RT-qPCR assay but negative by the RT-LAMP assay may be the very low copy numbers (having high CT values of >38 in the RT-qPCR assay). However, regarding the other two samples tested positive by the RT-LAMP assay (with a high time threshold of >45 min) but negative by the RT-qPCR assay, the reason is speculated to be the presence of viral variants, that have caused mismatches with the primers and/or probe [11,14].

Although the SARS-CoV-2 detection rate in nasal swabs was reported to be lower than in bronchoalveolar lavage fluid (BALF) and sputum, the high viral load in nasal swabs, combined with easy sampling, will facilitate the early diagnosis of COVID-19, especially for mild and asymptomatic infections [20]. In view of a mean viral load of 1.4 × 10^6^ copies/mL in nasal swabs of COVID-19 patients, the novel assay is sufficiently sensitive for the detection of SARS-CoV-2 at an early stage of infection using nasal swab specimens. Recently, the nucleic acid extraction-free protocol of the LAMP assay was developed by directly adding NaOH-treated swabs into the reaction [25]. The RNA extraction-free RT-LAMP assay for SARS-CoV-2 should be deployed in the future.

In brief, the optimal RT-LAMP assay for real-time monitoring is a 25 µL reaction mixture, containing: 1x isothermal amplification buffer; 6 mM MgSO4; 1.4 mM dNTPs; 8 units of WarmStart Bst 3.0 DNA polymerase; 7.5 units of WarmStartR RT; 0.15 unit of Q5 High-Fidelity DNA Polymerase; 1.6 μM each of primers FIP and BIP; 0.2 μM each of primers F3 and B3; 0.4 μM of loop primer LB; and 0.4 mM SYTO9. The optimized colorimetric reaction mixture contains: 1x WarmStart Colorimetric LAMP buffer; 0.15 unit of Q5 High-Fidelity DNA Polymerase; 1.6 μM each of primers FIP and BIP; 0.2 μM each of primers F3 and B3; and 0.4 μM of loop primer LB, in 25 µL volume reaction mixture. The optimized reaction temperature is 63 °C.

## 4. Materials and Methods

### 4.1. RNA Extraction

Viral RNA was extracted from 300 μL throat swabs of suspected COVID-19 patients, using an RNA extraction kit (Liferiver, Shanghai, China), and eluted in 90 μL of nuclease-free water for immediate use or storage at −80 °C. For the specificity tests, total nucleic acids were extracted from 200 μL throat swabs that tested positive for common respiratory viruses and virus culture supernatants (HCoV-OC43: VR-1558 and HCoV-229E:VR-740), using the MagNA Pure LC Total Nucleic Acid Isolation Kit (Roche Diagnostics GmbH, Mannheim, Germany) according to the manufacturer’s instruction, and then eluted in 100 μL of nuclease-free water.

### 4.2. Reaction System of the Novel RT-LAMP Assay

A 25 µL RT-LAMP reaction system was set up, containing: 1x isothermal amplification buffer; 6 mM MgSO4; 1.4 mM dNTPsl; 8 units of WarmStart Bst 3.0 DNA polymerase; 7.5 units of WarmStartR RT; 0.15 unit of Q5 High-Fidelity DNA Polymerase; 1.6 μM each of internal primers FIP and BIP; 0.2 μM each of outer primers F3 and B3; 0.4 μM of loop primer LB; and 0.4 mM SYTO9 (Life technologies, Carlsbad, CA, United States). Three kinds of polymerase were all purchased from New England Biolabs (Beverly, MA, United States). Three μL of RNA standard or template was added into each RT-LAMP reaction. The reaction was performed at 63 °C for 50 min by the LightCycler 96 real-time PCR System (Roche Diagnostics, Mannheim, Germany) for real-time monitoring.

### 4.3. Specificity of the Novel RT-LAMP Assay

The specificity of the SARS-CoV-2 RT-LAMP assay was evaluated using 17 common respiratory viruses. Two human coronaviruses (hCoVs), HCoV-OC43 (VR-1558) and HCoV-229E, were from standard strains of VR-1558 and VR-740, respectively, which were purchased from the American Type Culture Collection (ATCC). Nucleic acids of another 15 viruses (including: influenza A, B, and C viruses; parainfluenza viruses type 1–3; enterovirus; RSV A and B groups; HCoV-HKU-1; HCoV-NL63; human rhinovirus; human metapneumovirus; adenovirus; and bocavirus) were obtained from positive clinical samples from children with acute respiratory tract infections.

### 4.4. Sensitivity and Limit of Detection (LOD)

To prepare the RNA standard, a SARS-CoV-2 *N* gene fragment (28774-28971 nt in Wuhan-Hu-1, GenBank: MN908947.3) was amplified from a positive clinical sample with primers containing the T7 promoter. The RNA standard was obtained via in vitro transcription, and quantified by a Qubit^®^ 4.0 Fluorometer (Thermo Fisher Scientific, USA). The copy number of the RNA standard was calculated using the following formula: RNA copies/mL = [RNA concentration (g/μL)/(nt transcript length × 340)] × 6.022 × 10^23^.

Ten-fold serial dilutions of the RNA standard, from 10^5^ to 10^0^ copies per 1 μL, were used as the standards to determine the sensitivity of the novel SARS-CoV-2 RT-LAMP assay. For the LOD test, ten-fold serial dilutions of the RNA standard, from 10^4^ to 10^0^ copies per 1 μL, were used. Each dilution was tested in a set of 10 replicates. To more accurately estimate the LOD, additional experiments were performed by adding five-fold dilutions of the RNA standard (1,963 copies, 393 copies, 79 copies, 16 copies and 3.1 copies) to the 25 μL reaction volume. The LOD was defined as a 95% probability of obtaining a positive result, using probit regression analysis with SPSS 17.0 software [26].

### 4.5. Evaluation of the Novel SARS-CoV-2 Detection Assay Using Clinical Samples

To evaluate the performance of the novel RT-LAMP assay for SARS-CoV-2 detection, 56 throat swabs were collected from suspected COVID-19 patients and individuals who were admitted or quarantined at Nantong Third Hospital. SARS-CoV-2 infection was detected with a commercial SARS-CoV-2 RT-qPCR kit (Liferiver Bio, Shanghai, China) and the novel RT-LAMP assay. Because the amounts of RNA from clinical samples were very limited, we only performed a comparison between the real-time monitoring RT-LAMP assay and the commercial RT-qPCR assay. The concordance rate between both assays was calculated by the formula: (number of consistent results by both methods/total number) ×100%.

### 4.6. Visual Detection

For visual detection, a WarmStart Colorimetric LAMP 2X Master Mix (New England Biolabs, Beverly, MA, United States), that uses cresol red as a visual indicator, was used. A 25 µL reaction system was set up, containing: 1x WarmStart Colorimetric LAMP buffer; 0.15 unit of Q5 High-Fidelity DNA Polymerase; 1.6 μM each of primers FIP and BIP; 0.2 μM each of primers F3 and B3; and 0.4 μM of loop primer LB. The reactions were performed at 63 °C, and the color change from red to yellow was observed at the 30, 40, 50 and 60 min time points.

### 4.7. Ethics Statement

The study was approved by Nantong Third Hospital Ethics Committee (E2020002: 3 February 2020). Written informed consents were obtained from each of the involved patients.

## 5. Conclusions

In conclusion, we developed a new, simple and sensitive RT-LAMP assay for the fast and accurate detection of SARS-CoV-2. The RT-LAMP assay has a high sensitivity, with a LOD of 118.6 copies of SARS-CoV-2 RNA per 25 μL reaction, and good specificity regarding common respiratory viruses. Validation using 56 clinical samples confirmed that the RT-LAMP assay’s performance was similar to that of the RT-qPCR assay in the detection of SARS-CoV-2. Although their specificity regarding SARS-CoV and MERS-CoV was not tested, in view of the lethal nature of these two coronaviruses and SARS-CoV-2, a positive result for any of these three coronaviruses might be of clinical importance. The novel SARS-CoV-2 RT-LAMP assay has been developed into real-time monitoring and colorimetric versions, both of which will be especially useful in COVID-19 surveillance.

## Figures and Tables

**Figure 1 ijms-21-02826-f001:**
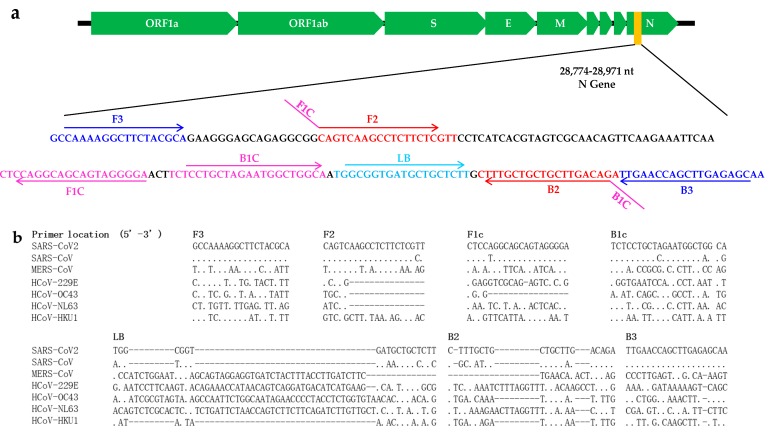
Primer information. (**a**) Location of the primers in SARS-CoV-2 genome; (**b**) Sequence comparison among seven human coronaviruses (SARS-CoV-2, SARS-CoV, MERS-CoV, OC43, HKU1, NL63 and 229E).

**Figure 2 ijms-21-02826-f002:**
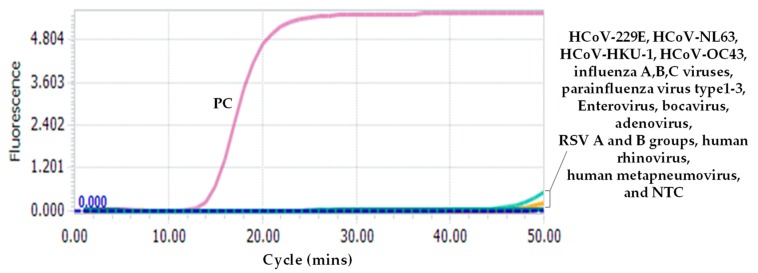
Specificity of the novel SARS-CoV-2 RT-LAMP assay. The common respiratory viruses used in this assay include: HCoV-HKU-1; HCoV-NL63; HCoV-OC43; HCoV-229E; influenza A, B, and C viruses; parainfluenza virus type 1–3; enterovirus; RSV A and B groups; human rhinovirus; human metapneumovirus; adenovirus; and Bocavirus. SARS-CoV-2 RNA standard was used as positive control (PC). NTC: non-template control.

**Figure 3 ijms-21-02826-f003:**
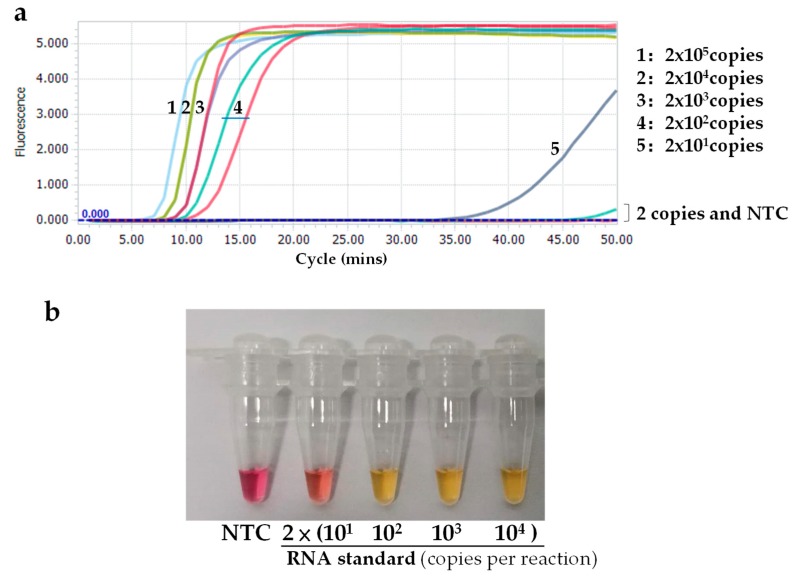
Sensitivity of the novel SARS-CoV-2 RT-LAMP assay. (**a**) Real-time monitoring with SYTO9; (**b**) Visual detection with cresol red. NTC: non-template control.

**Table 1 ijms-21-02826-t001:** Primer information of the novel SARS-CoV-2 RT-LAMP assay.

Primers	Sequence (5’-3’)	Length (nt)
F3	GCCAAAAGGCTTCTACGCA	19
B3	TTGCTCTCAAGCTGGTTCAA	20
FIP	TCCCCTACTGCTGCCTGGAGCAGTCAAGCCTCTTCTCGTT	40
BIP	TCTCCTGCTAGAATGGCTGGCATCTGTCAAGCAGCAGCAAAG	42
LB	TGGCGGTGATGCTGCTCTT	19

**Table 2 ijms-21-02826-t002:** Limit of detection (LOD) of the novel SARS-CoV-2 RT-LAMP assay.

Dilution	Standard (Copies/Reaction)	Positive/Total Tested
1×	1963	10/10
5×	393	10/10
5×	79	8/10
5×	16	6/10
5×	3	2/10
LOD	118.6 copies/reaction	

**Table 3 ijms-21-02826-t003:** Comparison of the novel RT-LAMP assay with a commercial RT-qPCR assay.

RT-qPCR Assay	The novel RT-LAMP Assay	Total	Positive Rate (%)	Concordance Rate (%)
Positive	Negative
Positive	34	2	36	64.3%	92.9%
negative	2	18	20
Total	36	20	56		
Positive rate (%)	63.4%

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
