# Peer review of "A Novel Reverse Transcription Loop-Mediated Isothermal Amplification Method for Rapid Detection of SARS-CoV-2"

_ijms, 2020, doi:10.3390/ijms21082826_

Round 1

Reviewer 1 Report

The presented report is striking relevance and should merit fast publication due the actual pandemic context. Still, there are some issues that should be addressed to sustain scientific robustness and not misleading readers.

1- The abstract should be more focused, presenting with any doubt the main findings of the report – the method and the output in terms of performance.

1.1- sentences referring to future applications and or claims not backed up by data should be removed – e.g. “… in primary care and/or community hospitals without the need for sophisticated equipment or skilled personnel” should be removed, since anyone working with LAMP based methods knows of the issues related to false positives.

1.2- “… when the template input is more than the LOD” – should be rephrased to clearly indicate the LOD in terms of comparison to existing validated methods at the frontline.

2- The same goes for the Conclusion – POCT is far from reality and not backed up by data. Please refrain from highlighting common generalizations about LAMP based methods.  

2.1- the whole Conclusion section should be edited to highlight the findings and not future perspectives. Summarize the main objective findings in terms of detection capability and performance.

2.2- In conclusions, the fact that the method is not selective for CoV2, but “Because corresponding clinical samples or standard strains are unavailable for SARS-CoV and MERS-CoV, we did not test the specificity for the two coronaviruses” – this is not a “bad” result. Since, even though not selective, if positive for any of these two other viruses, a positive result is clinical relevant – this should be highlighted in conclusions.

3- Experimental

3.1- please make sure you determine exact copy number of target for calibration curve from actual copy number attained by RT-qPCR. Estimation from overall concentration via spectroscopy methods present error of great magnitude.

3.2- please ensure that (perhaps at the end of discussion) you propose a “final” optimized protocol.  

3.3- make sure you present all data for different times (as you do) but also, the curves at “the optimal protocol”, i.e. cycle time

3.4- there are not data supporting (line 175) “an even patients’ own houses, regardless of whether it is in developed countries or developing counties, by directly adding NaOH-treated swabs into the reaction.” – that was not validated.

3.5- it is not clear form the way the report is drafted whether both the RNA samples from the clinical swabs and for the standard templates were treated the same way, i.e. it seems the kits were not the same and it would be interesting to know/validate any difference..

Reviewer 2 Report

This is clearly a timely paper introducing a useful novel assay for SARS-CoV-2 detection. However in places the authors have allowed their enthusiasm to run ahead of the data presented. 

In the Abstract (end) they suggest that skilled personnel are not required to perform the assay but such suggestion neglects the fact that this is a highly infectious nature of the virus which require s considerable care in handling such unqualified statements are somewhat irresponsible.

On line 80 a SARS-CoV-2 RNA standard is mentioned but the source and nature of this standard does not seem to be defined in the paper.

On the same line (80)  I presume 104 is a typo and should read 104.

In section 2.4 an evaluation using clinical material is described but the samples tested are not defined. This is important as later (lines 169-171) we are told nasal swabs testing results in lower rates of detection than BAL or sputum samples. It is not clear whether the data refers to this assay or an assay described in the incomplete reference 20 (refs 19 and 21 are similarly incomplete).

The comparison - section 2.4 - explains the 2 false negative results obtained with new assay but the two apparent false positive results are not discussed, were these false negative PCR results? How high was the signal in the loop amp assay?

Lines 174 -177 it is suggested the loop amp assay can be used anywhere. However unless I have missed something critical, all the samples tested using this developemtn had nucleic acid extraction prior to testing. Are the authors seriously suggesting that primary care centres and even domestic dweelings will have a Roche Magnapure and Liferiver extraction kits available? The final part of the sentence (line 177) suggests NaOh treatment of swabs – presumably an alternative to the high technology extraction they used – could be used. How can this statement be made when there is no data to support it?

In view of this the conclusions lines 235-241 are not supported by the data presented in the paper. The section should be removed or else rewritten to better reflect the data presented in the paper

Author Response

This is clearly a timely paper introducing a useful novel assay for SARS-CoV-2 detection. However in places the authors have allowed their enthusiasm to run ahead of the data presented. 

In the Abstract (end) they suggest that skilled personnel are not required to perform the assay but such suggestion neglects the fact that this is a highly infectious nature of the virus which require s considerable care in handling such unqualified statements are somewhat irresponsible.

On line 80 a SARS-CoV-2 RNA standard is mentioned but the source and nature of this standard does not seem to be defined in the paper.

Response:We thank the reviewer for this careful remark. We revised this part and removed the descriptions on the use in primary care and/or community hospitals by unskilled personnel”. For the SARS-CoV-2 RNA standard (mentioned on line 80), we gave detailed methodological description in section 4.4 (lines 215-227).

On the same line (80)  I presume 104 is a typo and should read 104.

Response:Yes, thank you for pointing out this error. It was corrected in the revised manuscript.

In section 2.4 an evaluation using clinical material is described but the samples tested are not defined. This is important as later (lines 169-171) we are told nasal swabs testing results in lower rates of detection than BAL or sputum samples. It is not clear whether the data refers to this assay or an assay described in the incomplete reference 20 (refs 19 and 21 are similarly incomplete).

Response:We provided sample information in section 4.5 (lines 229-236). The samples were throat swabs collected from suspected COVID-19 patients and individuals who were admitted or quarantined at Nantong Third Hospital.

For the comparison of viral load between different tissue samples, the result was directly cited from a previous publication (ref 20), and was not obtained by this study. With respect to the incomplete information of some references, the reason is that these literatures were published online ahead of print. We updated all cited references except for some that are still not published officially.

The comparison - section 2.4 - explains the 2 false negative results obtained with new assay but the two apparent false positive results are not discussed, were these false negative PCR results? How high was the signal in the loop amp assay?

Response:The possible reason for these two samples tested positive by the RT-LAMP assay (having high time threshold of >45 minutes) but negative by the RT-qPCR assay is speculated to be the presence of viral variants that had caused mismatches with primers and/or probe. We added this explanation in the discussion of the revised manuscript.

Lines 174 -177 it is suggested the loop amp assay can be used anywhere. However unless I have missed something critical, all the samples tested using this development had nucleic acid extraction prior to testing. Are the authors seriously suggesting that primary care centres and even domestic dwellings will have a Roche Magnapure and Liferiver extraction kits available? The final part of the sentence (line 177) suggests NaOh treatment of swabs – presumably an alternative to the high technology extraction they used – could be used. How can this statement be made when there is no data to support it?

Response:We thank the reviewer for careful reading of our paper. Indeed, we used extracted RNA for the assay. In the future, we will upgrade the assay to a direct isothermal amplification system using clinical samples. To be cautious, as suggested by the reviewer, we have removed the description on the potential use of our assay in in primary care and/or community hospitals by unskilled personnel.

In view of this the conclusions lines 235-241 are not supported by the data presented in the paper. The section should be removed or else rewritten to better reflect the data presented in the paper

Response:We rewrote this section according to the valuable comments.

Reviewer 3 Report

Review report about manuscript of Lu et al entitled: „A Novel Reverse Transcription Loop-Mediated Isothermal Amplification Method for Rapid Detection of SARS-CoV2”

COVID-19 is a threatening pandemic emerged last year, but its fast spread will affect the whole year everywhere of the globe. We learn more and more about the virus SARS-CoV2, the causative agent of the disease and now everybody is aware of that increasing the number of tests for its diagnosis is one clue for slow down the spread. Until now several qRT-PCR diagnostic tools were developed, but these test needs equipped persons and more importantly expensive equipment. Now the virus spreads in Europe and US where PCR machines are quite frequently available and even at this part of the world increasing the test capacity is a wish.

RT-LAMP is a straightforward solution for this demand. As the sequence of the virus is available, and it shows very low mutation rate it can be used for optimal primer designs what is usually the bottleneck of LAMP primer design. Moreover, Zhang and co-workers have recently published their   results on a mismatch -tolerant RT-LAMP method, what can even increase the sensitivity and reproducibility of these tests.

In this paper they summarized their new results using this RT-LAMP method to diagnose SARS-CoV2 with high sensitivity in RNA samples purified from throat swabs. Using available sequences of several human infecting coronaviruses, including SARS-CoV2 they designed primers for LAMP which were proved to be sensitive for the diagnosis of the virus when compared to “traditional” qRT-PCR tests.

I am convinced that the low number of discrepancies between the two methods can be explained by the low abundance of the virus in the samples, which did not correlate by the two methods. I think this paper is very important to be published very fast – this would allow production of commercial kits for its use.

If the test will be used as a routine test, it would be nice to include an inner control, to prove if the RNA extraction was successful – I missed this question here, however as this paper concentrate on the test (primers, and reaction mixtures of the RT-LAMP assay), it is not an absolute need here.

In the discussion possibility about RNA extraction free protocol is also envisioned, which would be a very important direction of the RT-LAMP. So, I would encourage the authors, that as a next step try this simple “crude extract” protocol, as if the pandemic enriches the non-developed part of the globe this simple, really on-site, test would help to save several lives in the future.

minor correction:

line 124: SARS-CoV2 has high affinity to ACE2 – or please specify than which SARS it has higher affinity

AS a summary I think that this paper is a nice work, very important to be published as soon as possible and I encourage the authors to continue their research and I would be pleased to review their next manuscript about the crude extract RT-LAMP of SARS-CoV2.

Author Response

Review report about manuscript of Lu et al entitled: „A Novel Reverse Transcription Loop-Mediated Isothermal Amplification Method for Rapid Detection of SARS-CoV2”

COVID-19 is a threatening pandemic emerged last year, but its fast spread will affect the whole year everywhere of the globe. We learn more and more about the virus SARS-CoV2, the causative agent of the disease and now everybody is aware of that increasing the number of tests for its diagnosis is one clue for slow down the spread. Until now several qRT-PCR diagnostic tools were developed, but these test needs equipped persons and more importantly expensive equipment. Now the virus spreads in Europe and US where PCR machines are quite frequently available and even at this part of the world increasing the test capacity is a wish.

RT-LAMP is a straightforward solution for this demand. As the sequence of the virus is available, and it shows very low mutation rate it can be used for optimal primer designs what is usually the bottleneck of LAMP primer design. Moreover, Zhang and co-workers have recently published their   results on a mismatch -tolerant RT-LAMP method, what can even increase the sensitivity and reproducibility of these tests. In this paper they summarized their new results using this RT-LAMP method to diagnose SARS-CoV2 with high sensitivity in RNA samples purified from throat swabs. Using available sequences of several human infecting coronaviruses, including SARS-CoV2 they designed primers for LAMP which were proved to be sensitive for the diagnosis of the virus when compared to “traditional” qRT-PCR tests.

I am convinced that the low number of discrepancies between the two methods can be explained by the low abundance of the virus in the samples, which did not correlate by the two methods. I think this paper is very important to be published very fast – this would allow production of commercial kits for its use.

Response:We thank the reviewer for the positive comments.

If the test will be used as a routine test, it would be nice to include an inner control, to prove if the RNA extraction was successful – I missed this question here, however as this paper concentrate on the test (primers, and reaction mixtures of the RT-LAMP assay), it is not an absolute need here.

Response:We thank the reviewer for this important suggestion. Inner control was often used in the RT-qPCR assay as a quality control of RNA extraction and reaction system because it can be detected with a different florescent channel from the target gene.

Distinct from qPCR assay, LAMP amplification was monitored by double-stranded DNA-binding fluorescent dye (e.g. SYBR Green I or SYTO 9), or pH-sensitive indicator dye (e.g. cresol red). Therefore, the inner control cannot be used in the same reaction tube. However, we agree that an inner control can be designed as a quality control for RNA extraction and reaction system in an independent tube.

In the discussion possibility about RNA extraction free protocol is also envisioned, which would be a very important direction of the RT-LAMP. So, I would encourage the authors,  that as a next step try this simple “crude extract” protocol, as if the pandemic enriches the non-developed part of the globe this simple, really on-site, test would help to save several lives in the future.

minor correction:

Response:Sure, we will develop the RNA extraction free protocol.  

line 124: SARS-CoV2 has high affinity to ACE2 – or please specify than which SARS it has higher affinity

Response:Thank the reviewer for pointing out this error. We corrected this sentence (it should be “higher than SARS-CoV”).  

AS a summary I think that this paper is a nice work, very important to be published as soon as possible and I encourage the authors to continue their research and I would be pleased to review their next manuscript about the crude extract RT-LAMP of SARS-CoV2.

Response:We appreciate the reviewer’s comment, and we will further develop the RNA extraction free protocol for easy use.

Round 2

Reviewer 1 Report

Please check line 98-99 “RT-qPCR”, I believe should be “RT-LAMP”

Author Response

Response to Reviewer 1 Comments

Please check line 98-99 “RT-qPCR”, I believe should be “RT-LAMP”

Response:Yes, thank you very much for pointing out this error. It was corrected in the revised manuscript.